# Different In Vitro-Generated MUTZ-3-Derived Dendritic Cell Types Secrete Dexosomes with Distinct Phenotypes and Antigen Presentation Potencies

**DOI:** 10.3390/ijms23158362

**Published:** 2022-07-28

**Authors:** Takuya Sakamoto, Terutsugu Koya, Misa Togi, Kenichi Yoshida, Tomohisa Kato, Yasuhito Ishigaki, Shigetaka Shimodaira

**Affiliations:** 1Department of Regenerative Medicine, Kanazawa Medical University, Kahoku 920-0293, Ishikawa, Japan; taku0731@kanazawa-med.ac.jp (T.S.); koya@kanazawa-med.ac.jp (T.K.); m-togi@kanazawa-med.ac.jp (M.T.); 2Center for Regenerative Medicine, Kanazawa Medical University Hospital, Kahoku 920-0293, Ishikawa, Japan; ken1-y@kanazawa-med.ac.jp; 3Medical Research Institute, Kanazawa Medical University, Kahoku 920-0293, Ishikawa, Japan; tkato@kanazawa-med.ac.jp (T.K.J.); ishigaki@kanazawa-med.ac.jp (Y.I.)

**Keywords:** dexosome, MUTZ-3, dendritic cells, antigen presentation

## Abstract

Human dendritic cell (DC) dexosomes were evaluated for their function and preclinical validation for vaccines. Dexosomes are small DC-secreted vesicles that contain absorbing immune signals. Vaccine manufacturing requires a significant number of monocyte-derived DCs (Mo-DCs) from donor blood; thus, Mo-DC dexosomes are expected to serve as novel materials for cancer vaccination. In this study, we characterized a potential dexosome model using immature and mature MUTZ3-derived DCs (M-imIL-4-DC, M-imIFN-DC, M-mIL-4-DC, and M-mIFN-DC) and their dexosomes (M-imIL-4-Dex, M-imIFN-Dex, M-mIL4-Dex, and M-mIFN-Dex). Despite the lack of significant differences in viability, M-mIFN-DC showed a significantly higher level of yield and higher levels of maturation surface markers, such as CD86 and HLA-ABC, than M-mIL-4-DC. In addition, M-mIFN-Dex expressed a higher level of markers, such as HLA-ABC, than M-mIL-4-Dex. Furthermore, M-mIFN-Dex exhibited a higher level of antigen presentation potency, as evaluated using a MART-1 system, than either M-imIFN-Dex or M-mIL-4-Dex. We found that M-mIFN-Dex is one of the four types of MUTZ3-derived DCs that harbor potential immunogenicity, suggesting that DC dexosomes could be useful resources in cancer immunotherapy.

## 1. Introduction

Dendritic cells (DCs) are antigen-presenting cells (APCs) that play a central role in the immune response targeting tumor-associated antigens (TAAs) [1]. DCs take up antigens and migrate into the lymph nodes, where the antigens are presented to naive T cells through major histocompatibility complexes on the DCs. DC-primed naive CD8^+^ T cells become cytotoxic T lymphocytes (CTLs) that attack tumors carrying TAAs [2,3]. DCs detected at lower frequencies, i.e., less than 2%, in the human peripheral blood mononuclear cells cannot be expanded in vitro [4,5]. Harvesting a large population of mononuclear cell-rich fractions is required either from patients or donors for manufacturing DC vaccines. Therefore, cancer vaccines using DCs are usually prepared by ex vivo monocyte differentiation isolated via apheresis [6,7,8]. However, DC vaccines are more susceptible to immunosuppressive molecule effects and signals in the tumor microenvironment, making long-term DC vaccine storage difficult while maintaining their expected efficacy as cell-based products [9,10]. Therefore, vaccine research has recently focused on cell-free DC exosomes (dexosomes) [11]. Dexosomes are small DC-secreted vesicles that contain absorbing immune signals. Dexosomes have been applied as cell-free anticancer vaccine materials in several clinical trials, including two Phase I [12,13] trials and one Phase II [14] trial in patients with end-stage cancer, due to their superior capacity as immunotherapeutic agents. Dexosomes are 30–150 nm in diameter and originate from multivesicular bodies (MVBs) in the endosomal pathway in activated DCs. Although dexosomes secreted by both immature and mature DCs exhibit typical exosomal characteristics, mature DC dexosomes display a higher capability to induce T cell immune responses [15]. TAA-bearing dexosomes on HLA class I and II in activated DCs deliver functional peptide–HLA complexes to other naive DCs. Paracrine-targeted DCs are then primed, and they acquire the ability to activate naive T cells that proliferate into tumor-derived peptide-specific CTLs [16]. In addition, mature DC dexosomes forming functional peptide–HLA complexes directly induce T cell expansion. Laurence et al. reported that dexosomes pulsed with tumor-derived peptides induced an anti-tumor T cell response leading to anti-tumor immunity in mice. They proposed that human dexosomes could be beneficial vectors as antigen delivery materials for cancer immunotherapy [17]. The use of dexosomes isolated from cultured DCs is expected to be a solution for several technical challenges associated with DC-based vaccines. Due to the antigen-presenting ability of DCs, the stability of the exosome membrane allows for cryopreservation for at least 6 months [18]. However, generating dexosomes from human monocyte-derived DCs requires a large number of DCs with inter-donor variability. Therefore, a potential strategy is to isolate a large amount of dexosomes using cell line-derived DCs.

Among multiple DC differentiation model cell lines, MUTZ-3, an acute myelomonocytic leukemia cell line, has been investigated as a beneficial model to generate human DCs in vitro [19]. MUTZ-3 progenitor cells can be differentiated into IL-4-DC and IFN-DC stimulated with GM-CSF, TNF-α, mitoxantrone, and either IL-4 or IFN-α, similar to the differentiation of peripheral monocytes into DCs [20]. Therefore, MUTZ-3-derived DCs are convenient alternative models to human monocytes in the peripheral blood for the generation of dexosomes from human DC-like cells. An additional advantage is that this large-scale culture allows a significant quantity of dexosomes with expected functions for further studies to be obtained. The principal aim of this study was to generate two types of MUTZ-3-DCs under conditions, including either IL-4 or IFN-α in a serum-free medium, and investigate the superiority of the dexosomes they secreted based on phenotype and functional analyses. The dexosomes from MUTZ-3-derived DCs could potentially provide useful tools for the development of a clinical platform using dexosomes in cancer immunotherapy.

## 2. Results

### 2.1. MUTZ-3-Derived IL-4-DC and IFN-DC Generation Using a Serum-Free Medium

Fetal bovine serum (FBS) is required for the generation of MUTZ-3-derived DCs, and the vesicles in FBS might affect the dexosome-related experimental results. Hence, we established a protocol for MUTZ3-derived DC generation using a serum-free medium. To isolate the exosomes secreted by immature and mature MUTZ-3-derived DCs (dexosomes), MUTZ-3 progenitor cells were cultured in a serum-free medium supplemented with a cytokine cocktail for 4 days to generate immature MUTZ-3 derived DCs (M-imIL-4-DCs and M-imIFN-DCs), and either IL-4 or IFN-α mature MUTZ-3-derived DCs (M-mIL-4-DCs and M-mIFN-DCs) were generated through 24-h maturation (Figure 1). Dendritic-like morphology was observed using microscopy in the case of both M-mIL-4-DCs and M-mIFN-DCs (Figure 2a). The viabilities of both mature MUTZ-3-derived DC types were similarly close (72–78%) to the initial MUTZ-3 cell number (Figure 2b left; mean viability: M-mIL-4-DC, 71.8%; M-mIFN-DC, 78.3%). The yield of M-mIFN-DCs was significantly higher than that of M-mIL-4-DCs. (Figure 2b right; mean yield: M-mIL-4-DC, 24.6%; M-mIFN-DC, 35.0%).

M-imIL-4-, M-mIL4-, M-imIFN-, and M-mIFN-DCs were generated as described in the Materials and Methods. Dexosomes were prepared using culture supernatant in immature and mature MUTZ-3-derived DC. Dexosomes were isolated from the culture supernatant via elution from magnetic beads and phosphatidylserine binding protein (M-imIL-4Dex, M-mIL-4-Dex, M-imIFN-Dex, and M-mIFN-Dex).

### 2.2. DC Generation by IFN-α Increased Maturation Marker Expression on DCs

Next, we analyzed the expression levels of maturation markers in M-imIL-4-DC, M-im-IFN-DC, M-mIL-4-DC, and M-mIFN-DC. Obtaining the expression levels of surface molecules, such as CD80, CD86, CD83, HLA-ABC, HLA-DR, and CD40, was necessary to reveal the antigen presentation ability, which was higher in mature DCs and stimulated CD8^+^ T cells in vitro [21,22]. Both M-mIFN-DC and M-mIL-4-DC exhibited significantly higher levels of maturation markers than primary MUTZ-3, indicated by the percentage of positive cells and ΔMFI of CD80, CD86, CD83, HLA-ABC, and CD40 in Figure 3. These results indicated that MUTZ-3 could differentiate into DCs either with IL-4 or IFN-α stimulation (Figure 3 and Appendix A). The ΔMFI of CD80, CD86, CD83, HLA-ABC, and CD40 in M-mIFN-DC was significantly higher due to the process of maturation. The ΔMFI levels of various markers on M-imIFN- and M-mIFN-DCs were as follows: CD80, 72.3 and 228.5; CD86, 149.2 and 765.9; CD83, 28.05 and 591.8; HLA-ABC, 848.1 and 1916; and CD40, 546.8 and 2117, respectively. Furthermore, the ΔMFI of CD86 and HLA-ABC was significantly higher in M-mIFN-DC than that in M-mIL-4-DC (for M-mIL-4- and M-mIFN-DCs, respectively: CD86, 278 and 765.9; HLA-ABC, 1265 and 1916).

### 2.3. Characterization of MUTZ-3-Derived IL-4- and IFN-DCs That Secrete Dexosomes

The dexosomes from the culture supernatant of immature and mature MUTZ-3-derived DCs were definitively isolated (Figure 1). Hereafter, the immature and mature M-IL-4-DC dexosomes and M-IFN-DC dexosomes are referred to as “M-imIL-4-Dex,” “M-imIFN-Dex,” “M-mIL-4-Dex,” and “M-mIFN-Dex,” respectively. Nanoparticle tracking analysis (NTA) of the vesicles revealed similar size distributions for M-imIL-4-Dex, M-imIFN-Dex, M-mIL-4-Dex, and M-mIFN-Dex. A comparison of mode diameters showed no significant difference in the sizes of M-mIL-4-Dex and M-mIFN-Dex. When dexosome numbers were quantified to normalize the DCs from which they were derived, we identified a significant increase in the number of M-mIL-4-Dex per DC through maturation (Figure 4a). To evaluate the potential presence of dexosomes, we determined the expression of exosome markers such as CD9, CD63, and CD81 using flow cytometry (Figure 4b and Appendix A). We observed lower CD9 and CD81 exosome marker levels in mature dexosomes than in immature dexosomes (mean percentage of positive cells: CD9, 96.5% and 37.1% and CD81, 99.8% and 96.9%, in M-imIL-4-Dex and M-mIL-4-Dex, respectively; CD9, 79.0% and 49.2%, and CD81, 99.2% and 98.9%, in M-imIFN-Dex and M-mIFN-Dex, respectively. ΔMFI: CD9, 4.3 and 1.8, and CD81, 758.4 and 248.5, in M-imIL-4-Dex and M-mIL-4-Dex, respectively; CD9, 3.1 and 2.5, and CD81, 736.9 and 326.0, in M-imIFN-Dex and M-mIFN-Dex, respectively).

### 2.4. MUTZ-3-Derived IFN-DC Dexosomes Exhibit High Expression of HLA-ABC

As the comparative phenotype analyses of MUTZ3-derived IL-4- and IFN-induced DCs indicated higher expression of maturation markers (Figure 3), we evaluated whether similar findings could be observed in MUTZ-3-derived IL-4- and IFN-DC dexosomes. As shown in Figure 5 and Appendix A, the positive percentage and ΔMFI of HLA-ABC in M-imIFN-Dex and M-mIFN-Dex were significantly higher than those in M-imIL-4-Dex and M-mIL-4-Dex (mean percentage of positive cells of HLA-ABC, 2.6% and 44.2%; ΔMFI of HLA-ABC, 0.1 and 1.7 in M-imIL-4-Dex and M-imIFN-Dex, respectively; mean percentage of positive cells of HLA-ABC, 7.8% and 49.6%; ΔMFI of HLA-ABC, 0.3 and 2.3 in M-mIL-4-Dex and M-mIFN-Dex, respectively). The other DC maturation markers did not show significant differences between M-IL-4-Dex and M-IFN-Dex. Therefore, distinct dexosomes with different phenotypes could be provided by the roots of MUTZ-3-derived DCs.

### 2.5. MUTZ-3-Derived IFN-DC Dexosomes Exhibit Higher Antigen-Presenting Ability

To investigate the antigen presentation ability to CD8^+^ T cells in isolated dexosomes, MUTZ-3-derived IL-4- and IFN-DC dexosomes were loaded directly with the MART-1 peptide. MART-1 specific CD8^+^ T cells could be detected using MART-1 tetramer analysis on days 14 and 21. The M-mIFN-Dex showed the number of MART-1 tetramer^+^/CD8^+^ positive cells with a time-dependent increase on days 14 and 21 compared with M-mIL-4-Dex in six healthy donors (Figure 6a,b). Furthermore, the levels of MART-1 tetramer^+^/CD8^+^ positive cells in M-mIFN-Dex were significantly higher through the process of maturation than those in M-im-IFN-Dex. This quantitative analysis indicated that M-mIFN-Dex induced higher numbers to present antigens to CD8^+^ T cells than M-mIL-4-Dex.

## 3. Discussion

In this study, we evaluated the functions of dexosomes secreted by MUTZ-3-derived IL-4- and IFN-DCs. The viability, yield, and DC phenotypes in MUTZ-3-derived IL-4- and IFN-DCs are considered critical attributes for dexosome quality verification. Despite the lack of significant differences in viability, M-mIFN-DC exhibited a significantly higher level of yield than M-mIL-4-DC. Furthermore, M-mIFN-DC expressed significantly higher levels of maturation markers, such as CD86 and HLA-ABC, than M-mIL-4-DC. In addition, M-imIFN-Dex and M-mIFN-Dex expressed higher levels of HLA class I than M-imIL-4-Dex and M-mIL-4-Dex. Moreover, the M-mIFN-Dex exhibited a potentially higher level of antigen presentation capability based on the MART-1 peptide through HLA class I.

Traditionally, human monocyte-derived DCs have been used to study DC biology and DC vaccination efficacy, in which CD14^+^ monocytes are relatively abundant in the peripheral blood. To reveal the characteristics of dexosomes in vitro requires harvesting a significant number of monocytes from the peripheral blood. Hence, an experimental model using a cell line is a durable resource for the mechanistic evaluation of DC functions. Santegoets et al. reported that tumor cells of myeloid and lymphoid lineages exhibit the potential to differentiate into DC-like APCs. Furthermore, this DC-differentiating potential has also been shown for leukemia-derived cell lines, especially those that originate from myelogenous or monocytic lineages, such as THP-1, HL-60, KG-1, K562, and MUTZ-3. However, other than MUTZ-3, these cells have low differentiation potential into DCs [19]. The phenotypic and functional characteristics of MUTZ-3 derived DCs that were essential for CTL-mediated immune response generation in vivo [23,24]. Furthermore, Jurjen et al. reported that MUTZ-3-derived IL-4-DCs and IFN-DCs established a model for studying DC functions in vitro [20]. Studies have shown that IFN-α drives the rapid differentiation of monocytes into highly activated DCs exhibiting the characteristics of functional APCs [25,26]. Delayed intracellular proteolysis, efficient routing to recycling vesicles, and long-term antigen presentation ability are specific IFN-DC features [27]. Given that the experiments using MUTZ-3-derived DCs require FBS, there was a concern that the vesicles in FBS would affect the dexosome-related experimental results [28]. Therefore, we used a serum-free medium in our protocol for generating MUTZ3-derived DCs with IL-4 and IFN-α to evaluate the characteristics of dexosomes (Figure 1).

M-mIFN-DC and M-mIL-4-DC exhibited significantly increased expression of maturation markers, such as CD80, CD86, CD83, HLA-ABC, and CD40, compared with MUTZ-3, confirming that MUTZ-3 differentiated into DCs (Figure 3 and Appendix A). Moreover, M-mIFN-DC displayed a significant increase in the expression levels of mature markers (CD86 and HLA-ABC) and yield compared with those of M-mIL-4-DCs (Figure 2 and Figure 3). Mature DCs express cell surface co-molecules necessary for antigen presentation, such as CD80, CD86, CD83, and CD40, as well as HLA-ABC and HLA-DR, with higher expression levels than those of immature DCs commitment with antigens to T cells in vitro [29,30]. We previously reported that maturation markers on DCs were clinically presumed to detect antigen-specific CTLs induced by a DC vaccine with a potential antigen-presenting ability [31]. Therefore, it could be suggested that higher levels of maturation markers on DCs provide superior antigen presentation ability, leading to immune acquisition. As shown in Appendix A, the M-mIFN-DCs tended to show higher antigen-presenting ability induction than M-mIL-4-DCs. The present results suggested that M-mIFN-DCs might have a superior antigen-presenting ability. IFN-α is known to rapidly differentiate human monocytes into DCs known as IFN-DCs, and it is effective in terms of mediating cross-priming and cross-presentation of CD8^+^ T cells [27]. Further in vivo studies will be required to clarify whether M-mIFN-DCs have an antigen-presenting ability superior to that of M-mIL-4-DCs. TLR (Toll-like receptor) signaling that strongly induces maturation and survival of human DCs was reported to have no influence on DC maturation in MUTZ-3 [32]. Therefore, the sensitivity of MUTZ-3- and monocyte-derived DCs would be different. The same might be elucidated for the characteristics of MUTZ-3-derived dexosomes. Further studies are needed to determine whether MUTZ-3- and monocyte-derived DC-dexosomes would be differentially characterized.

DC-derived extracellular vesicles that induce exosomes and microvesicles are part of a rapidly developing research field with applications that include diagnostics and therapeutics. DC exosomes not only carry DC-presented antigens, but also contain functional MHC–peptide complexes [16,33]. The characteristics of dexosomes released from MUTZ-3-derived IL-4- and IFN-DCs have never been evaluated. The amount of dexosomes obtained from single cell secretion increased in M-mIL-4-DCs and M-mIFN-DCs, and both M-mIL-4-Dex and M-mIFN-Dex displayed typical exosomal characteristics (Figure 4 and Appendix A). The mean size of mature M-IFN-Dex is only slightly larger (111.5 nm) than that of M-mIL-4-Dex (100.6 nm), and the same amount of dexosome secretion could be detected from a single cell, suggesting that dexosome size and amount did not determine different functional dexosome profiles. Sara et al. reported that the phenotype of human monocyte-derived DCs is partly reflected in exosomes [34], and their results are consistent with studies in mice conducted by Segura [35]. We evaluated whether this phenomenon is observed in MUTZ-3-derived IL-4- and IFN-DC dexosomes. The ΔMFI of CD86^+^ or CD40^+^ cells were higher in M-mIFN-DC than that in M-mIL-4-DC (Figure 3), although the percentage of positive exosomes and ΔMFI in CD86^+^ and CD40^+^ were not increased in M-mIFN-Dex (Figure 5). We could not detect the MUTZ3-derived IL-4- and IFN-DC surface molecules CD80 on the dexosomes (Figure 5 and Appendix A). Furthermore, the expression levels of CD80, CD86, CD83, HLA-ABC, and CD40 increased significantly with the maturation procedure in M-mIFN-DC, whereas these findings were not observed in dexosomes. Our results demonstrated that the phenotypes of MUTZ-3-derived DCs were reflected in exosomes for some but not all molecules. Although the mechanism of exosome biosynthesis is not understood entirely, it is generally accepted that exosomes are formed and developed through endocytic pathways, such as early endosomes and late endosomes, and subsequently released into the extracellular environment through exocytosis [36]. Spadaro et al. analyzed the subcellular distribution of MHC-I in IFN-DC and IL-4-DC [27]. They reported that MHC-I molecules in IFN-DCs were preferentially located in early endosomes involved in exosome formation. These reports suggest that HLA-ABC of HLA class I may be preferentially located in M-mIFN-Dex and not M-mIL-4-Dex during exosome formation. Further studies will be required to clarify whether HLA-ABC in M-mIFN-DC, compared with in M-mIL-4-DC, are preferentially located in the early endosomes involved in exosome formation. CD81, CD86, and CD40 have been reported to interact with MHC molecules in APCs to regulate their antigen-presenting capacities [37,38]. However, the expression of CD81 in mature MUTZ-3-DC dexosomes was lower than that in immature MUTZ-3-DC dexosomes, and did not change between MUTZ-3 derived IL-4- and IFN-DC dexosomes. Furthermore, CD86 and CD40 levels were unchanged between MUTZ-3-IL-4- and IFN-DC-dexosomes. Therefore, these results suggest that CD81, CD86, and CD40 on the dexosomes were not involved in the antigen-presenting ability.

Casper et al. reported that ovalbumin-loaded dexosomes were superior to microvesicles in terms of inducing antigen-specific CD8^+^ T cells [39]. Zitvogel et al. also demonstrated that dexosomes gave a more effective tumor suppression response than parental DCs, and that self-dexosomes could induce antigen specific CTLs, highlighting the role of dexosomal MHC class I in ex vivo processes [17]. Another study showed that human dexosomes directly loaded with MART-1 peptide possessed functional peptide–MHC complexes focusing on DCs in vitro [40]. We have shown that HLA class I on dexosomes can directly load the MART-1 peptides that stimulate CD8^+^ T cells (Appendix A). Compared with M-mIL-4-Dex, M-mIFN-Dex was more efficient in inducing antigen-presenting ability immune responses (Figure 6a,b). The increased levels of HLA-ABC among HLA-associated antigens on M-mIFN-Dex could be partially associated with MART-1-specific induction of antigen presentation observed in all healthy donors. Despite no significant difference in HLA-ABC expression between M-imIFN-Dex and M-mIFN-Dex, there was a significant difference in the MART-1-specific induction of antigen-presenting ability at day 21. The difference in induction of antigen-presenting ability between M-imIFN-Dex and M-mIFN-Dex may be due to changes in the protein composition and priming abilities of dexosomes, which reflect the maturation signals received by DCs. Mature DC dexosomes contain higher levels of intercellular adhesion molecule 1 than are found in immature DC dexosomes, but bear low levels of milk fat globule-epidermal growth factor VIII (MFG-E8) compared with those in immature DC dexosomes, which may allow mature DC dexosomes to expose targeted cells more efficiently than immature DC dexosomes [35]. Therefore, the active protein composition and priming abilities of dexosomes might be reflected in the mature profiles with DCs. Further studies will be required to clarify whether M-mIFN-Dex contains ICAM-1 and MFG-E8.

The differentiated ratio of MUTZ-3-DCs indicated less than approximately 40% of the seeded MUTZ-3 cells, presuming that extensive cell death occurred in these cultures during differentiation (Figure 2b). Apoptotic cells release apoptotic bodies of 500 nm to 2 µm after they are degraded into intracellular fragments [41]. The M-IL-4- and M-IFN-DC-derived dexosome size was observed to be approximately 100 nm (Figure 4a), suggesting that the dexosomes isolated with magnetic beads were unlikely to be contained within apoptotic bodies. Furthermore, Ramesh et al. reported that apoptotic cells released exosomes. However, the underlying mechanism of apoptotic cell-derived exosome release is poorly understood and has only begun to be elucidated [42]. Further studies are warranted to determine how apoptotic cell-derived dexosomes affect the antigen-presenting capacity.

Our results revealed that treatment with IFN-α enhances the functional profiles of MUTZ-3-derived DCs, and that the highly expressed HLA-ABC-positive dexosomes they secrete have high antigen-presenting abilities. Furthermore, dexosome characteristics may enable large-scale and long-term M-IFN-Dex preservation, while maintaining efficacy. Accordingly, M-IFN-DC dexosomes might be efficient vehicles for HLA-ABC-peptide complexes in antigen presentation, enabling potent T cell activation. Based on our results, M-mIFN-DC dexosomes, with their higher antigen-presenting capability, could be useful for improving efficacy as treatment resources. Our data suggest that antigen-presenting molecules of HLA-ABC on M-IFN-Dex exist; thus, it is expected that M-IFN-Dex could function as a cancer vaccine, effectively carrying tumor antigen peptides. In addition, experimental models of MUTZ-3-derived DC dexosomes could be useful tools in the functional analyses of cell-free anticancer vaccines.

## 4. Materials and Methods

### 4.1. Subjects and Ethics Statement

This study was approved by the Ethics Committee of Kanazawa Medical University (approval numbers: G131 and I489). All cellular materials were obtained from healthy donors after they provided written informed consent. Peripheral blood samples were collected from healthy donors using blood collection tubes (BD Biosciences, Franklin Lakes, NJ, USA), and the centrifuged mononuclear cells were washed with PBS. All investigations were performed in accordance with the declaration of Helsinki.

### 4.2. MUTZ-3-Derived DC Generation

MUTZ-3 cells (Deutsche Sammlung von Mikroorganismen und Zellkulturen [DSMZ], Braunschweig, Germany) were maintained by seeding 4 × 10^5^ cells every three days in MEM-α medium (Gibco, Life Technologies Co., Carlsbad, CA, USA), supplemented with 20% FBS (Gibco), 5 ng/mL human recombinant GM-CSF (Peprotech, Rotterdam, The Netherlands). MUTZ-3-derived IL-4- and IFN-DCs were induced by culturing MUTZ-3 cells (3 × 10^5^ cells/mL) in AIM-V medium (serum-free medium, Thermo Fisher Scientific, Inc., Waltham, MA, USA), supplemented with 250 ng/mL GM-CSF (Peprotech, The Netherlands), 12 ng/mL TNF-α (Peprotech), 2nM Mitoxantrone (Sigma-Aldrich, Zwijndrecht, The Netherlands), and either 10 ng/mL IL-4 (Miltenyi Biotec B.V. & Co. KG, Bergisch Gladbach, Germany) or 5 ng/mL IFN-α2b (Peprotech) for the induction of immature MUTZ-3-derived IL-4- and IFN-DCs. After 3 days, the M-imIL4-DC or M-imIFN-DC were harvested, counted, and maturated by seeding 3 × 10^5^ cells/mL M-imIL4-DC or M-imIFN-DC in AIM-V medium, supplemented with 120 ng/mL TNF-α (Peprotech), 0.75 ng/mL IL-1β (Peprotech), and 1 μg/mL PGE2 (Kyowa Pharma Chemical Co., Ltd., Toyama, Japan). Either M-mIL4-DC or M-mIFN-DC were harvested after 24 h and used for subsequent experiments. Immature and mature MUTZ-3-derived IL-4- and IFN-DCs are referred to as “M-imIL-4-DC”, “M-imIFN-DC”, “M-mIL-4-DC”, and “M-mIFN-DC”, respectively.

### 4.3. MUTZ-3-Derived IL-4- and IFN-DC-Dexosome Preparation

The culture supernatant from either MUTZ-3-derived IL-4- or IFN-DCs were collected and subjected to three successive centrifugations: 300× *g* for 5 min to remove the cells; 1200× *g* for 20 min to remove the debris; and 10,000× *g* for 30 min to remove the large EVs. The culture supernatant was collected, and dexosomes were isolated using the MagCapture exosome isolation kit (Fujifilm Wako Pure Chemical Co., Osaka, Japan) according to the manufacturer’s instructions.

### 4.4. Morphological Cell Analysis

Harvested M-mIL-4-DCs and M-mIFN-DCs were observed using fluorescence microscopy (EVOS^®^ FL Cell Imaging System; Thermo Fisher Scientific, Inc.).

### 4.5. Cell Surface Marker Analysis of MUTZ-3-Derived DCs

To examine the expression of cell surface markers on MUTZ-3-derived IL-4- and IFN-DCs under each condition, aliquots of 1 × 10^5^ DCs were prepared in the FACSFlow ^TM^ (BD Biosciences). DCs were treated with human FcR Blocking Reagent (Miltenyi Biotec B.V. & Co. KG) for 10 min at room temperature. Each aliquot was incubated with the following mouse IgG anti-human monoclonal antibodies conjugated to fluorescein isothiocyanate (FITC), phycoerythrin (PE), and allophycocyanin (APC): FITC-conjugated anti-CD80 mAbs (BD Biosciences), PE-conjugated anti-CD86 mAbs (eBioscience, Inc., San Diego, CA, USA), APC-conjugated anti-CD83 mAbs (BioLegend, Inc., San Diego, CA, USA), FITC-conjugated anti-CD40 mAbs (eBioscience, Inc.), FITC-conjugated anti-HLA-ABC mAbs (BD Biosciences), and PE-conjugated anti-HLA-DR mAbs (eBioscience, Inc.). The antibodies were added to the cells and incubated for 30 min at 4 °C in the dark. After incubation, the cells were washed with FACSFlowTM (BD Biosciences) and centrifuged at 500× *g* and 4 °C for 5 min. DCs were resuspended in FACSFlowTM containing 7-amino-actinomycin D (7-AAD; BD Biosciences) to detect dead cells. The live cells, defined as negative for 7-AAD, were gated on forward scatter (FSC) and side scatter (SSC) in the main population (Appendix A). Gated cells were examined for immunophenotyping. All analyses were performed on a flow cytometer (FACS Calibur, Becton Dickinson, San Jose, CA, USA), and the data were analyzed with the Flowjo software (BD Biosciences).

### 4.6. Phenotypic Dexosome Characterization

To examine the surface marker expression on the dexosomes, dexosomes in the culture supernatant from MUTZ-3-derived IL-4- and IFN-DCs were isolated using a PS Capture^TM^ Exosome Flow Cytometry Kit (Fujifilm Wako Pure Chemical Co., Osaka, Japan) according to the company’s protocol, followed by a flow cytometric analysis of surface antigens after immunostaining with fluorescence-labeled antibodies. Immature and mature MUTZ-3-derived IL-4- and IFN-DC dexosomes are referred to as “M-imIL-4-Dex”, “M-imIFN-Dex”, “M-mIL-4-Dex”, and “M-mIFN-Dex”, respectively.

### 4.7. NTA

Dexosome size measurement was conducted using the NanoSight NS300 instrument (Malvern, UK), following the manufacturer’s protocol. The instrument uses a laser light source to illuminate nanoscale particles (10–1000 nm), seen as individual point-scatters moving under Brownian motion. Each sample was diluted with PBS (80-fold dilution) and analyzed in five 60-s recordings using camera level 15, detection threshold 1, the auto minimum expected particle size, and auto jump distance in NTA version 3.0, then quantified and measured.

### 4.8. Direct Loading of MART-1 Peptide on Dexosomes

To directly load HLA-A*02:01 melanoma antigen recognized by T cells 1 (MART-1) peptides (ELAGIGILTV; synthesized by GeneScript, Nanjing, China) on MUTZ-3-derived IL-4- and IFN-DC dexosomes, 100 μL of purified dexosomes were mixed with an equal volume of 0.2 mol/L sodium acetate (pH 4.2) with 100 μg/mL of MART-1 peptide, and incubated at room temperature for 30 min. The dexosome preparation mixture was neutralized to pH 7.0 with 2 mol/L Tris-HCl (2.6% of total volume) of pH 11. Unbound MART-1 peptides were removed by filtering through a 100-kDa cut-off Amicon Ultra centrifugal Filter Unit (Millipore, Bedford, MA, USA).

### 4.9. Detection of Antigen Presentation In Vitro

Human leukocyte antigen (HLA) haplotypes from MUTZ-3 cells were determined by molecular HLA typing. The HLA type of MUTZ-3 is HLA-A*02:01 (Appendix A). As mentioned above, the direct loading of MART-1 peptide on dexosomes from either MUTZ-3-derived IL-4- or IFN-DCs were collected as stimulators. CD8^+^ T cells were separated from HLA-A*02:01-autologous PBMCs by using CD8 microbeads (Miltenyi Biotec B.V. & Co. KG) and were applied as responder CD8^+^ T cells. Stimulator and responder cells were cocultured in a 96-well U-bottom plate at a ratio of 2000 (dexosomes):1 (CD8^+^ T cells) or 1 (DCs):10 (CD8^+^ T cells) in AIM-V medium (Thermo Fisher Scientific, Inc.) supplemented with 5 ng/mL of IL-2 (PeproTech, Inc., Rocky Hill, NJ, USA), 5 ng/mL of IL-7 (Research and Diagnostic Systems, Inc., Minneapolis, MN, USA), 10 ng/mL of IL-15 (PeproTech, Inc., Rocky Hill, NJ, USA), and 50 µM of 2-mercapto-ethanol (Bio-Rad La-boratories, Inc., Hercules, CA, USA) as a stimulation medium. After 3 days of cultivation, AIM-V media supplemented with 5% human AB serum (Biowest, Nuaillé, France) and 50 µM 2-mercapto-ethanol were added as expansion media. Thereafter, dexosome stimulation and cell expansion were repeated twice with a 3-day interval. Cocultured cells were collected 14 and 21 days after the first stimulation, and 1 × 10^6^ cells were stained with FITC-conjugated anti-CD8 (Beckman Coulter, Inc., Brea, PA, USA), APC-conjugated anti-CD3 (eBioscience, Inc., San Diego, CA, USA), and PE-conjugated T-Select HLA-A*02:01 MART-1 Tetramer-ELAGIGILTV (Medical & Biological Laboratories Co., Ltd., Nagoya, Japan) to detect MART-1-specific CD8^+^ T cells. Dead cells were excluded by 7-AAD staining through flow cytometric analysis.

### 4.10. Statistical Analysis

We used a paired two-tailed Student’s t-test, one-way analysis of variance followed by Tukey’s multiple comparison test, and a Wilcoxon signed-rank test to compare the differences among the groups. All statistical analyses were performed using GraphPad Prism (version 9; GraphPad Software Inc., San Francisco, CA, USA). Differences were considered statistically significant at *p*-values of *p* < 0.05.

## 5. Conclusions

In conclusion, we showed that dexosomes secreted by various cytokine-induced and MUTZ3 cell line-differentiated DCs harbor functional DC characteristics. M-mIFN-DC showed higher maturation marker expression than M-mIL-4-DC. Furthermore, the HLA-ABC expression of M-mIFN-DC was reflected in the dexosomes. Compared with M-mIL-4-Dex, M-mIFN-Dex functionally exhibited the potency of antigen presentation to CD8^+^ T cells in vitro. M-mIFN-Dex is expected to be useful for developing dexosome vaccines and providing further insights into dexosomes in vitro and in vivo.

## Figures and Tables

**Figure 1 ijms-23-08362-f001:**
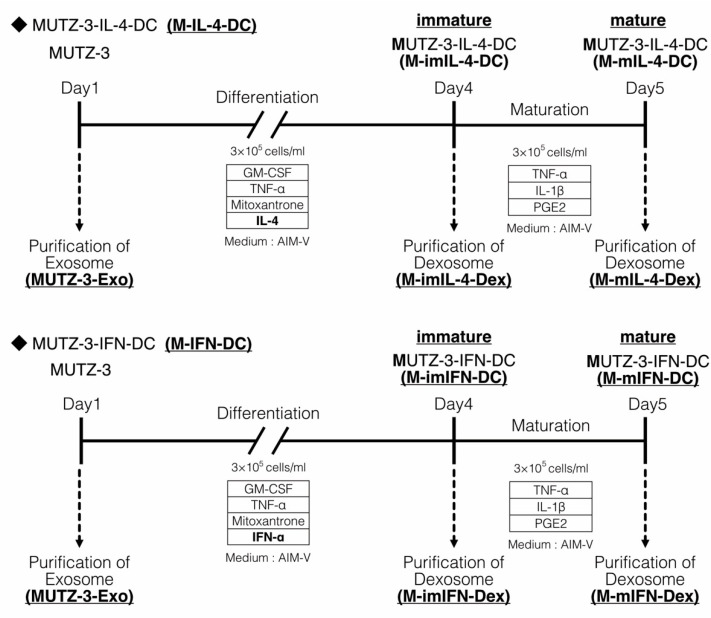
Protocol of MUTZ-3-derived Dendritic cell (DC) and dexosome induction.

**Figure 2 ijms-23-08362-f002:**
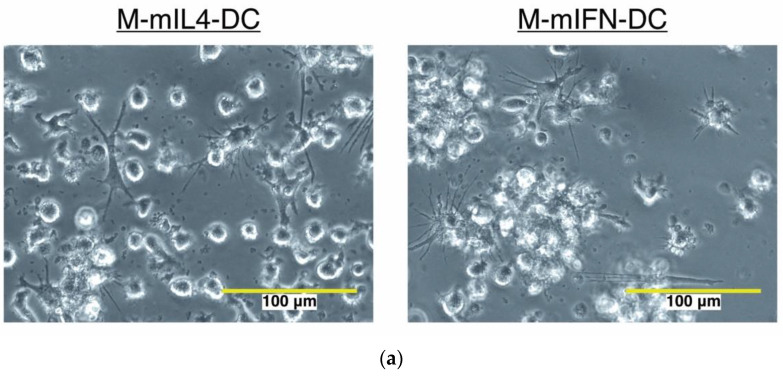
Comparison of cell morphology, cell viability, and yield in mature MUTZ-3-derived IL-4-DCs and IFN-DCs (M-mIL-4-DC and M-mIFN-DC). (**a**) Image of cells observed by phase-contrast microscopy before harvesting mature DCs. The white bar indicates 100 μm. (**b**) Live and dead cells were measured by trypan blue staining to compare the viability and yield of the MUTZ-3-derived DCs/seeded MUTZ-3 (*n* = 7). The bars indicate the mean ± S.D. of independent experiments. * *p* < 0.05 indicates a statistically significant difference compared to M-mIL-4-DC. The rhombus in the Scatter plot is MUTZ-3 derived IL-4-DCs. Circle is MUTZ-3 derived IFN-DCs.

**Figure 3 ijms-23-08362-f003:**
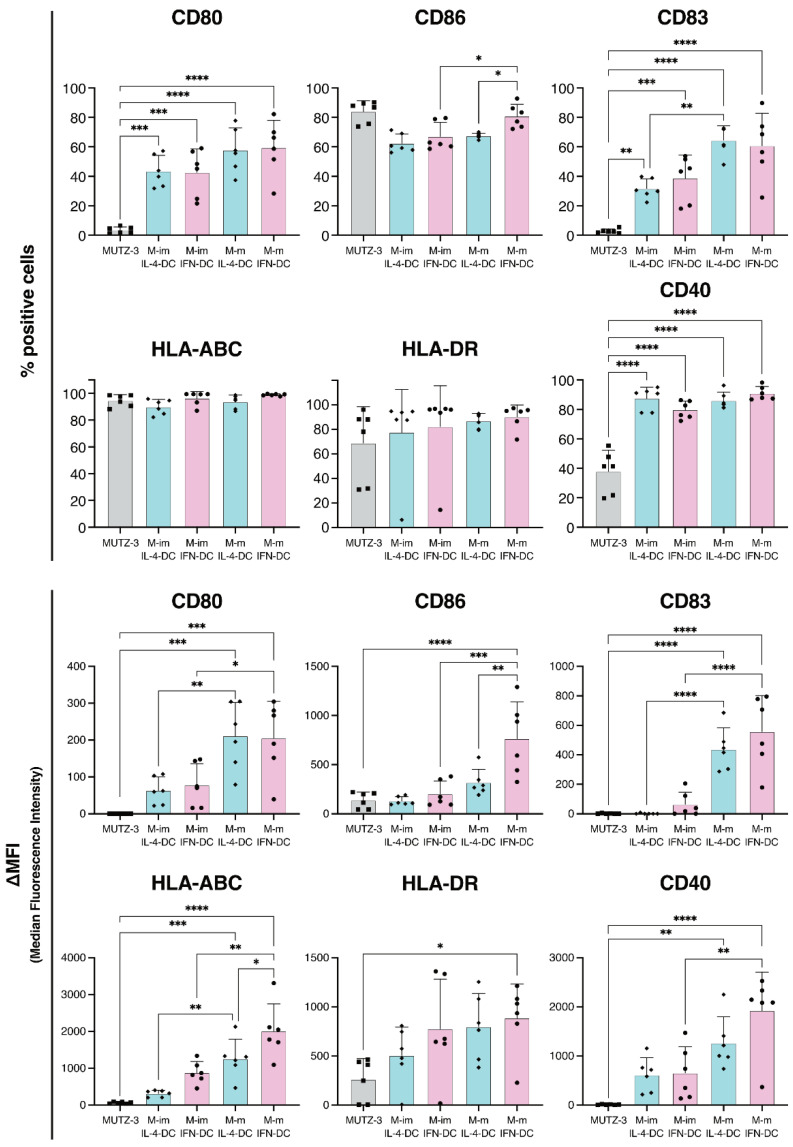
Comparison of immature and mature MUTZ3-derived IL-4-DCs and IFN-DCs phenotypes (M-imIL-4-DC, M-imIFN-DC, M-mIL-4-DC, and M-mIFN-DC). The results are shown as the percentage of positive cells and ΔMFI. The Δ median fluorescence intensity (ΔMFI) was calculated by subtracting the isotype control MFI values from the observed values. * *p* < 0.05, ** *p* < 0.01, *** *p* < 0.001, and **** *p* < 0.0001 indicate statistically significant differences (*n* = 6). The bars indicate the mean ± S.D. of independent experiments. The rhombus in the Scatter plot is MUTZ-3 derived IL-4-DCs. Circle is MUTZ-3 derived IFN-DCs. Square is MUTZ-3 cells.

**Figure 4 ijms-23-08362-f004:**
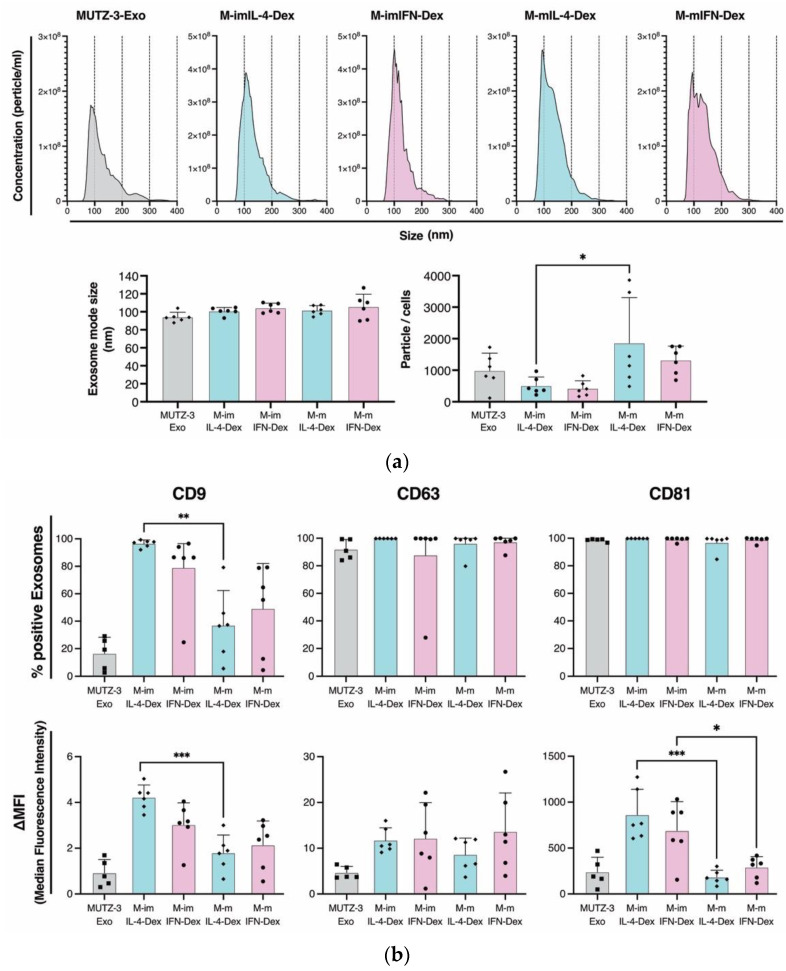
Characterization of MUTZ-3-derived IL-4- and IFN-DC dexosomes in immature and mature (M-imIL-4-Dex, M-imIFN-Dex, M-mIL-4-Dex, and M-mIFN-Dex, respectively). (**a**) The culture supernatants of immature and mature MUTZ-3 derived DCs were collected and the dexosomes were isolated. Size distribution, and the number of dexosomes secreted from cells identified by the nanoparticle tracking analysis (*n* = 6). (**b**) The dexosomes were stained with antibodies for exosome markers and analyzed using flow cytometry. The results are shown as a percentage of positive cells and ΔMFI. The Δ median fluorescence intensity (ΔMFI) was calculated by subtracting isotype control MFI values from observed values. * *p* < 0.05, ** *p* < 0.01, and *** *p* < 0.001 indicate statistically significant differences (*n* = 6). The bars indicate the mean ± S.D. of independent experiments. The rhombus in the Scatter plot is MUTZ-3 derived IL-4-DCs. Circle is MUTZ-3 derived IFN-DCs. Square is MUTZ-3 cells.

**Figure 5 ijms-23-08362-f005:**
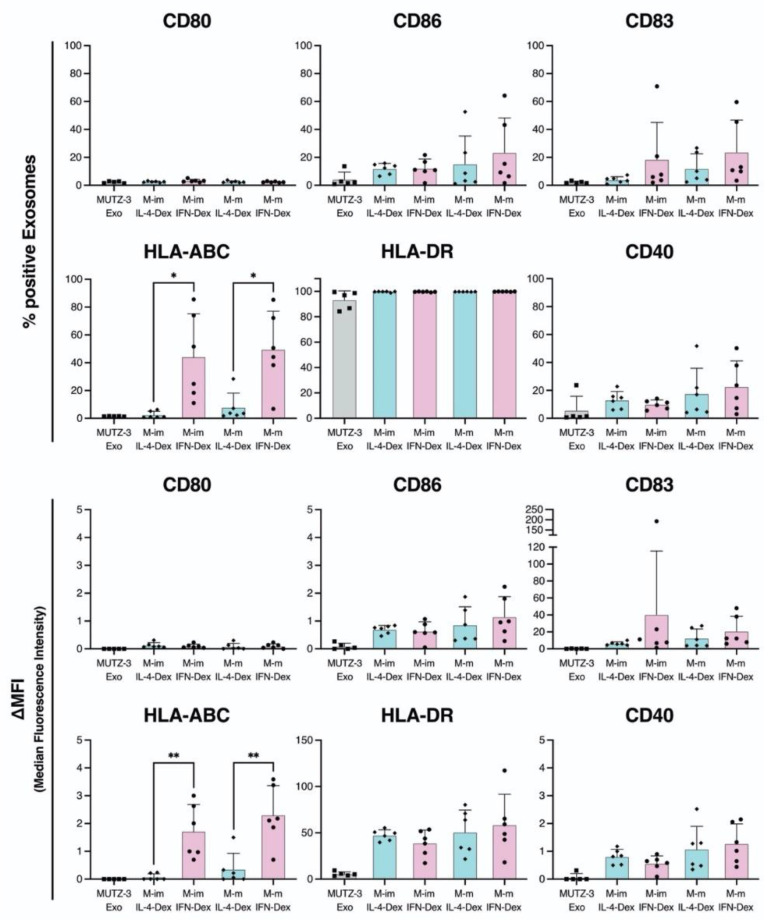
Comparison of immature and mature MUTZ-3-derived IL-4- and IFN-DC dexosome phenotypes. The DC maturation markers are shown as a percentage of positive exosomes and ΔMFI. The Δ median fluorescence intensity (ΔMFI) was calculated by subtracting isotype control MFI values from observed values. * *p* < 0.05 and ** *p* < 0.01 indicate statistically significant differences (*n* = 6). The bars indicate the mean ± S.D. of independent experiments. The rhombus in the Scatter plot is MUTZ-3 derived IL-4-DCs. Circle is MUTZ-3 derived IFN-DCs. Square is MUTZ-3 cells.

**Figure 6 ijms-23-08362-f006:**
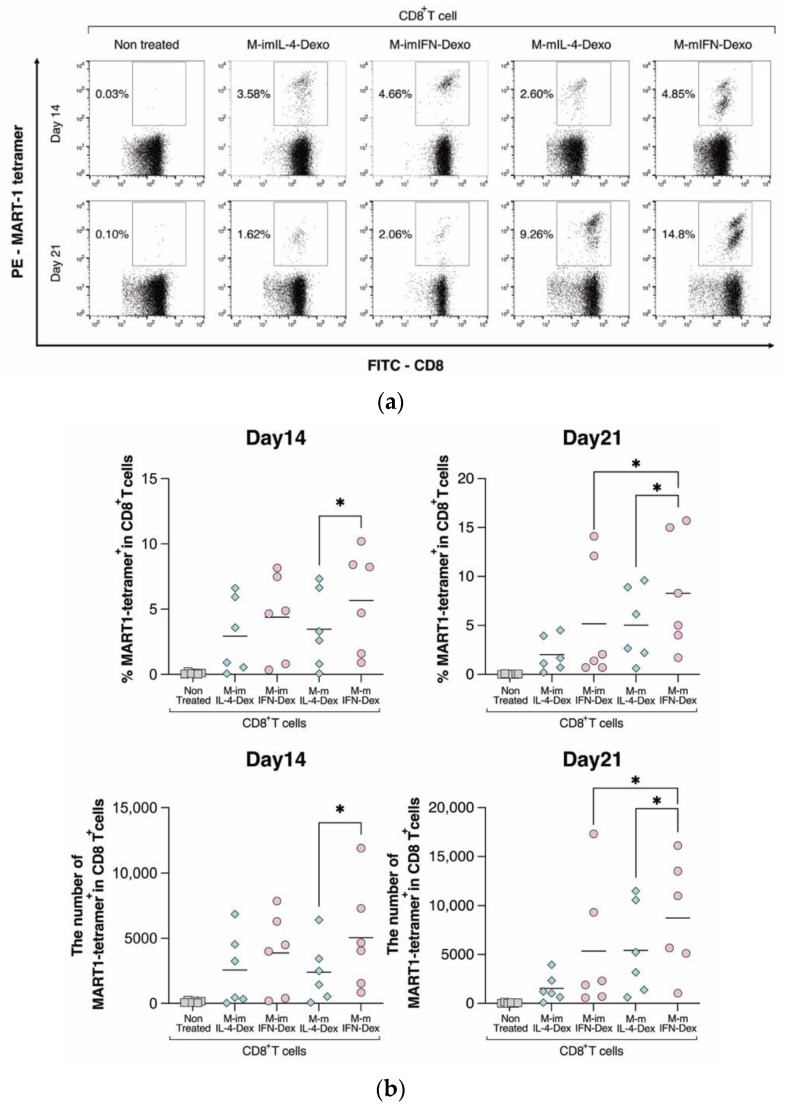
Comparison of MART-1-specific CD8^+^ T cell induction in immature and mature MUTZ-3-derived IL-4- and IFN-DC dexosomes. MUTZ-3-derived IL-4- and IFN-DC dexosomes were cocultured with autologous T cells at a ratio of 2000:1 (dexosomes:T cells). MART-1-specific CD8^+^ T cells were detected via CD3, CD8, and MART-1^+^ tetramers using flow cytometry 14 and 21 days after the start of the co-culture. (**a**) These dot plots show a representative example. The percentages in the panels indicate the MART-1 tetramer^+^ ratio in CD8^+^ T cells. (**b**) The graph shows the number of MART-1 tetramer^+^/CD8^+^ T cells in the culture period. The bold horizontal bars indicate the mean of each parameter. * *p* < 0.05 indicates a statistically significant difference (*n* = 6). The rhombus in the Scatter plot is MUTZ-3 derived IL-4-DCs. Circle is MUTZ-3 derived IFN-DCs. Square is MUTZ-3 cells.

## Data Availability

The data presented in this study are available in the article or Appendix A.

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
