# Peer review of "Different In Vitro-Generated MUTZ-3-Derived Dendritic Cell Types Secrete Dexosomes with Distinct Phenotypes and Antigen Presentation Potencies"

_ijms, 2022, doi:10.3390/ijms23158362_

Round 1

Reviewer 1 Report

The authors fulfilled all the reviewer's criticisms.

Please check on lane 295, it is not clear the meaning of the correction (CD6?). The sentence in lanes 229-234 should be revised. It was not easy to follow all the amendments performed by the authors, especially without a detailed reply and the big amount of tracked changes.

Author Response

We thank you greatly for your reply and review our manuscript (Manuscript ID: ijms-1780187). We appreciate the comments provided by the reviewer that have allowed us for further improvement of our manuscript. We have carefully revised the manuscript following the reviewer's suggestion. All changes have been made in a red character.

The authors fulfilled all the reviewer's criticisms.

Please check on lane 295, it is not clear the meaning of the correction (CD6?).

A.  As reviewer’s kind suggestion, we clearly revised this sentence in lines 244 of the revised manuscript.

The sentence in lanes 229-234 should be revised. It was not easy to follow all the amendments performed by the authors, especially without a detailed reply and the big amount of tracked changes.

A. As reviewer’s kind suggestion, we clearly revised this sentence in lines 232-235 of the revised manuscript.

Reviewer 2 Report

In their study Sakamoto and coworkers have characterized the immunophenotype of cultures of the human CD34+ leukemia cell line Mutz-3 in response to differential treatment (IL-4 versus IFN-alpha) and subsequent stimulation and of dexosomes derived from these cultures. Stimulated Mutz-DC and derived exosomes displayed largely comparable marker expression. However, in line with higher HLA-ABC levels, exosomes derived from stimulated IFN-alpha/Mutz-DC and loaded with a tumor-associated peptide induced more proliferation of cocultured CD8+ T cells derived form different donors.

The authors performed numerous in depth experiments with clear-cut results. However, several issues need to be addressed in addition:

The authors need to clarify that CD34+ Mutz-3 cells acquire a Langerhans cell-like phenotype (when applying mitoxantrone) and therefore may clearly differ from CD14+ monocyte-derived DC. The same may hold true for the characteristics of derived exosomes.

Figure 2: The finding that the numbers of differentially differentiated Mutz-3 cells were below approx. 40% of seeded cells suggests extensive cell death in those cultures in the course of differentiation. Viability was assessed by trypan blue staining. This technique does not allow to clearly discriminate (early) apoptotic cells. Therefore, the authors need to discuss potential influences of cell debris/apoptotic bodies on the behavior of the surviving cells. Further, the authors should discuss whether their exosome isolation technique may lead to unintended co-purification of apoptotic bodies.

Figure 6: the authors termed CD8+ T cells that were detectable by tetramer staining as CTL. However, usage of this term requires that according CD8+ T cells have been subjected to functional assays which demonstrated their cytotoxic activity. In case the authors can not provide additional FACS data to confirm the CD8+ T cell activation state (e.g. CD25 expression, intracellular cytokine detection) and have not performed CTL killing assays, they should not make use of this term.

Author Response

We thank you greatly for your reply and review our manuscript (Manuscript ID: ijms-1780187). We appreciate the comments provided by the reviewer that have allowed us for further improvement of our manuscript. We have carefully revised the manuscript following the reviewer's suggestion. All changes have been made in a red character.

In their study Sakamoto and coworkers have characterized the immunophenotype of cultures of the human CD34+ leukemia cell line Mutz-3 in response to differential treatment (IL-4 versus IFN-alpha) and subsequent stimulation and of dexosomes derived from these cultures. Stimulated Mutz-DC and derived exosomes displayed largely comparable marker expression. However, in line with higher HLA-ABC levels, exosomes derived from stimulated IFN-alpha/Mutz-DC and loaded with a tumor-associated peptide induced more proliferation of cocultured CD8+ T cells derived from different donors.

The authors performed numerous in-depth experiments with clear-cut results. However, several issues need to be addressed in addition:

The authors need to clarify that CD34+ Mutz-3 cells acquire a Langerhans cell-like phenotype (when applying mitoxantrone) and therefore may clearly differ from CD14+ monocyte-derived DC. The same may hold true for the characteristics of derived exosomes.

A. As reviewer’s kind suggestion, we added revision of sentence in lines 256-261 of the revised manuscript. And we added a new Reference 32 to the revised manuscript.

TLR (Toll-like receptor) signaling that strongly induces maturation and survival of human DCs reportedly unaffected MUTZ-3-derived DCs [32]. Therefore, the sensitivity of MUTZ-3- and monocyte-derived DCs would be different. The same might be elucidated for the characteristics of MUTZ-3-derived dexosomes. Further studies are needed to determine whether MUTZ-3- and monocyte-derived DC-dexosomes would be differentially characterized.

Figure 2: The finding that the numbers of differentially differentiated Mutz-3 cells were below approx. 40% of seeded cells suggests extensive cell death in those cultures in the course of differentiation. Viability was assessed by trypan blue staining. This technique does not allow to clearly discriminate (early) apoptotic cells. Therefore, the authors need to discuss potential influences of cell debris/apoptotic bodies on the behavior of the surviving cells. Further, the authors should discuss whether their exosome isolation technique may lead to unintended co-purification of apoptotic bodies.

A. As reviewer’s kind suggestion, we added revision of sentence in lines 326-336 of the revised manuscript. And we added a new Reference 41 and 42 to the revised manuscript.

The differentiated ratio of MUTZ-3-DCs indicated less than approximately 40% of the seeded MUTZ-3 cells, presuming that extensive cell death occurred in these cultures during differentiation (Figure 2b). Apoptotic cells release apoptotic bodies of 500 nm to 2 µm after they are degraded into intracellular fragments [41]. The M-IL-4- and M-IFN-DC-derived dexosome size was observed to be approximately 100 nm (Figure 4a), suggesting that the dexosomes isolated with magnetic beads were unlikely to be contained with apoptotic bodies. Furthermore, Ramesh et al. reported that apoptotic cells released exosomes. However, the underlying mechanism of apoptotic cell-derived exosome release is poorly understood and has only begun to be elucidated [42]. Further studies are warranted to determine how apoptotic cell-derived dexosomes affect the antigen-presenting capacity.

Figure 6: the authors termed CD8+ T cells that were detectable by tetramer staining as CTL. However, usage of this term requires that according CD8+ T cells have been subjected to functional assays which demonstrated their cytotoxic activity. In case the authors cannot provide additional FACS data to confirm the CD8+ T cell activation state (e.g. CD25 expression, intracellular cytokine detection) and have not performed CTL killing assays, they should not make use of this term.

A. As reviewer’s kind suggestion, we clearly revised this sentence in lines 186, 187-193, 198, 201, 204, 250-251, 309, 312, 314-315, 432, 452, 473, 476 of the revised manuscript. And we clearly revised this sentence in line 25, 29-30, 44-45, 50 of the supplemental figure manuscript.

This manuscript is a resubmission of an earlier submission. The following is a list of the peer review reports and author responses from that submission.

Round 1

Reviewer 1 Report

This manuscript by Sakamoto et al. describes a culture technique for the serum-free production of dexosomes from MUTZ-3 acute myelomonocytic leukemia cell line, a cell model previously described for the production of microvesculated particles for immunizatioin purposes. The authors describe a serum-free method of dexosome harvest from MUTZ-3 subjected to an IL-4 or IFNg stimulation protocol and demonstrate differential cytotoxic T cell activation upon dexasome MART-1 - HLA peptide complex.

The manuscript is well written and the experiments and conclusions are described logically. There are some concerns:

The reported consistent differences in the ability of IL-4 and IFNg dexosomes to elicit CTL responses coincides with IFNg dexosomes presenting more HLA which may enable these dexosomes to present MART-1 peptide more effectively to HLA-matched CTL. It also suggests that the IL-4 differentiation protocol described is inferior to the IFNg protocol. In fact, the two protocols only differ in the use of either IL-4 or IFNg, whereas all other ingredients remain constant (Fig. 1). Experience with in-vitro differentiation of monocytic cell models clearly show that the M1- or M2-like differentiation requires different cocktails of cytokines. Hence, the low HLA presentation (Fig. 5) and CTL activation (Fig. 6) shown here for dexosomes of MUTZ-3 cells treated with IL-4 may be explained by an inferior IL-4 differentiation protocol.

It is unclear why the authors did not try to optimize separate conditions for both IL-4 and IFNg dexosome production in MUTZ-3. This lack has a significant impact on the outcome of the study. While the authors may be able to comment on IFNg effects of dexosomes, it is questionable with respect to IL-4 yielded dexosomes and underscores the importance of an optimized initial differentiation protocol of MUTZ-3. The authors should not use identical diff protocols but rather tailor diff protocols to achieve equivalent HLA that may co-express with a  different balance of co-stimulatory factors and then measure those dexosomes for HLA-matched CTL responses. 

Reviewer 2 Report

In the present manuscript, the authors characterized two different cell models of mature DC starting from the myelomonocytic leukemia cell line  MUTZ-3 after the treatment with a cytokine cocktail containing IL-4 or IFNalpha. Exosomes purified from these cells were also characterized for membrane marker expression, loaded with MART-1 peptide, and used to activate CTL.

While the addressed item is very interesting, several concerns arose from the manuscript.

Dexosome stands for dendritic cell exosomes, this should be explained the first time and then the authors should refer to it as DC-exosomes or simply dexosomes.

As a general consideration, panels in the figures should be better identified (by using capital letters for example). This is particularly evident in fig 2 and 4. Moreover, supplementary figures should be also described in the results or in the materials and methods section, as opportune. Indeed figure S2 and S4 have been only described in the discussion while fig S3 does not seem to be described at all. Table 1 is lacking.

The graphical abstract is misleading. The highlighted increase in activation markers in M-IFN-DC refers to the comparison between M-IL4 and M-IFN-MUTZ-3 DC and not to immature MUTZ-3 cells. The same for dexosomes analyses.

In the introduction, (lanes 35-36) the sentence is not correct. Only naïve CD8+ T cells can become CTL.

In figure 3, as shown in figure S2, also the data of membrane markers from immature MUTZ-3 should be reported. This could help to better appreciate the meaning of the difference between two mature cells.

In the results, lanes 140-141, the meaning of the sentence is not clear. Please explain.

In the legends of figure 4, 5, and in the results (lane 157) exosomes have been referred to as cells, this is not correct.

Fig 5 is misleading since it is difficult to understand why, in the discussion, the authors declare that they do not detect CD40 molecules on dexosomes since the percentage of CD40 is similar to the one of CD83 or HLA-ABC. The difference is clearer in figure S2. To better clarify this point, staining of dexosomes from immature DC should be shown. This could help to understand the real meaning of the data.

The results presented in figure 6 are too preliminary and not convincing. No statistics have been performed. Maybe it could be useful to increase the number of observations. Moreover, to correctly assess the ability of M-IL4- or M-IFN-dexosomes, the right controls should be dexosomes from immature DC. This could allow to evaluate the biological meaning of the activation processes. It is also very difficult to understand how M-IL4-Dexo can promote (if true) CTL proliferation with such a low percentage of HLA antigens. Indeed, the differences in the % or MFI of costimulatory molecules between M-IL4 and M-IFN do not explain the differences (if true) in T cell activation among the three different tested donors. How did the authors assess the real load of MART-1 peptides on dexosomes?

In the discussion, the sentence in lanes 217-220 should be supported by statistically validated data. The same could be applied to lanes 226-228.

Data related to figure 4S should be presented in the results section and the partial association should be validated by a relevant statistical analysis.

In the materials and methods section, data from patients that underwent DC vaccination do not seem to be present in the manuscript (lanes 276-277).

Round 2

Reviewer 1 Report

The authors have addressed the concerns adequately. Minor editing of English language is encouraged. 

Author Response

We thank you greatly for your reply and review our manuscript (Manuscript ID: ijms-1633951).

We also appreciate the time and effort you and each of the reviewers have dedicated to providing insightful feedback on ways to strengthen our paper. 

We have been advised to revise the manuscript according to the reviewers’ comments, and the deadline for submission is April 14, 2022. 

However, as there are several comments need to be answered, I need two months to conduct the major revision. 

Therefore, we realized realize the submission delay will cause some inconveniences for you, we would be grateful if you could extend the deadline by 2 months to June 14, 2022. 

Your understanding and kind support will be appreciated.

Reviewer 2 Report

In the revised version of the manuscript, the authors coped with most of the reviewer’s concerns, and the resulting manuscript is surely ameliorated. However, the results presented in figure 6 are not yet acceptable.  If the authors want to demonstrate a superior capability of exosomes from M-IFN-DC to induce CTL activation, it is mandatory that results are shown together with relevant statistical analysis and using adequate controls. This is a very important part of the manuscript and qualitative results are not acceptable.

As suggested in the previous revision, dexosomes from immature MUTZ-3 cells, loaded with MART-1 peptides should be used as controls instead of untreated cells, and results of CTL proliferation from different donors should be validated by statistical analysis. Results presented in the new figure S4 and S5 are informative but they are only qualitative since, again, it lacks statistical significance.

If the authors want to demonstrate that exosomes from MUTZ-3 derived DC are able to activate CTL, independently from the way DC are treated, they should however use exosomes from untreated MUTZ-3 cells as control and validate their data with statistical analysis. In this case, adequate changes should be made in the abstract, result, and discussion sections and the title changed.

Finally, please insert also in the abstract the explanation of the meaning of “dexosomes”.

Author Response

(The authors gave the same response as above.)
